# CausalMamba: Scalable Conditional State Space Models for Neural Causal Inference

## Abstract

We introduce CausalMamba, a scalable framework that addresses fundamental limitations in fMRI-based causal inference: the ill-posed nature of inferring neural causality from hemodynamically-distorted BOLD signals and the computational intractability of existing methods like Dynamic Causal Modeling (DCM). Our approach decomposes this complex inverse problem into two tractable stages: BOLD deconvolution to recover latent neural activity, followed by causal graph inference using a novel Conditional Mamba architecture. On simulated data, CausalMamba achieves 37% higher accuracy than DCM. Critically, when applied to real task fMRI data, our method recovers well-established neural pathways with 88% fidelity, whereas conventional approaches fail to identify these canonical circuits in over 99% of subjects. Furthermore, our network analysis of working memory data reveals that the brain strategically shifts its primary causal hub—recruiting executive or salience networks depending on the stimulus—a sophisticated reconfiguration that remains undetected by traditional methods. This work provides neuroscientists with a practical tool for large-scale causal inference that captures both fundamental circuit motifs and flexible network dynamics underlying cognitive function.

## 1 Introduction

Inferring directed neural interactions from fMRI remains a crucial yet challenging problem. Because fMRI measures blood-oxygenation level dependent (BOLD) signals rather than neural activity itself, inference must proceed through a neurovascular forward model in which neural events are convolved with region- and subject-specific hemodynamic response functions (HRFs). This indirection introduces ambiguities and variability that complicate causal analysis, especially in the presence of feedback loops, non-stationarity, and latent confounds Das & Fiete (2020); Power et al. (2012); Aguirre et al. (1998); Handwerker et al. (2004); Krauth et al. (2022); Huang et al. (2019). The neuroscience field's standard approaches face fundamental limitations. DCM offers a biophysically principled framework capable of estimating flexible, region-specific HRF parameters Friston et al. (2003); Daunizeau et al. (2011). However, the computational complexity of Bayesian model inversion scales poorly to large networks. Despite advancements in optimization and faster variants (e.g., rDCM), this remains a significant bottleneck for whole-brain analysis. In contrast, data-driven methods such as Granger causality are computationally efficient but remain sensitive to hemodynamic distortions Granger (1969); Deshpande et al. (2010). To address these issues, we decompose the hemodynamic inverse problem into two tractable stages—differentiable BOLD deconvolution and causal graph inference—within a unified curriculum learning framework.

Modeling these neurovascular dynamics requires architectures that efficiently process long time-series while maintaining differentiability for end-to-end optimization. While Transformers face quadratic complexity, State Space Models (SSMs) offer a linear-time alternative well-suited for capturing hidden neural dynamics Valdes-Sosa et al. (2011); Huang et al. (2019). We introduce **CausalMamba**, a differentiable two-stage framework that incorporates a learnable hemodynamic component for HRF estimation and BOLD deconvolution, and subsequently infers directed causal connectivity from the deconfounded dynamics. At its core is **Conditional Mamba**, which integrates ROI-specific information into SSMs to capture both general temporal patterns and regional neurovascular heterogeneity. Our contributions are as follows:

- We introduce **CausalMamba**, a scalable framework that addresses the hemodynamic inverse problem by decomposing neural causal inference into two tractable stages: differentiable BOLD deconvolution and causal inference, achieving superior performance on both simulated and real fMRI data.

- We present a **Conditional Mamba architecture** that efficiently integrates ROI-specific information into state-space modeling, enabling the capture of both temporal dynamics and regional neurovascular heterogeneity in causal inference tasks.

- We demonstrate that our approach can recover well-established neural pathways in task fMRI data with substantially higher consistency than conventional methods (DCM, Granger Causality), highlighting its potential as a reliable tool for causal discovery in neuroscience.

## 2 RELATED WORK

**Neural Causality Discovery**    In neuroscience, Dynamic Causal Modeling (DCM) and Granger Causality (GC) remain standard frameworks for inferring neural connectivity Friston et al. (2003); Granger (1969). While DCM offers a biophysically principled approach by modeling the transformation from neural activity to hemodynamic signals, its application is often challenged by its sensitivity to model specifications (e.g., HRF assumptions) and prohibitive computational scaling with network size Daunizeau et al. (2011); Nag & Uludag (2024). Granger Causality, on the other hand, is limited by its sensitivity to hemodynamic confounds and reflects statistical association rather than true causation Yin & Barucca (2022); Barnett & Seth (2014). Although general-purpose causal discovery methods exist (e.g., NOTEARS Zheng et al. (2018), DAG-GNN Yu et al. (2019)), they are ill-suited for neuroimaging data as they fail to account for domain-specific challenges like indirect BOLD measurements and neurovascular heterogeneity.

**Conditional Modeling**    Conditional computation modulates network behavior using side information, from early methods like Conditional VAEs Kingma et al. (2014) and GANs Mirza & Osindero (2014) to more sophisticated hypernetworks Ha et al. (2016), adapters Houlsby et al. (2019), and FiLM Perez et al. (2018). Our Conditional Mamba extends FiLM-style modulation to state-space models with ROI-specific parameters, enabling both ROI-agnostic temporal modeling and targeted region-specific adaptation.

**State-Space Models**    SSMs are widely used for modeling hidden dynamics in both neuroscience (e.g., DCM) Valdes-Sosa et al. (2011) and causal discovery in nonstationary environments Huang et al. (2019). The Mamba architecture Gu & Dao (2023) augments SSMs with selective mechanisms for efficient, long-sequence modeling. We leverage this scalability to build a biophysically grounded, ROI-adaptive framework for neural causal inference.

## 3 METHODS

### 3.1 DATA CURATION

To rigorously train and validate CausalMamba, we curated a benchmark of synthetic and real-world data. We first generated a large-scale synthetic dataset with verifiable ground truth designed to replicate realistic neurovascular dynamics. Using a neural mass model, we produced LFP-like signals with biologically grounded connectivity, which were then convolved with varied, region-specific double-gamma HRFs to model inter-regional heterogeneity. The simulation incorporates realistic noise profiles (e.g., pink noise, motion artifacts), and the resulting BOLD signals were normalized and downsampled to fMRI resolution. For real-world validation, we used motor task fMRI time series from the Human Connectome Project (HCP). This dual-data approach provides a challenging and ecologically valid benchmark for evaluating causal inference methods (details in Appendices A, B)

## 3.2 Overall Model Framework

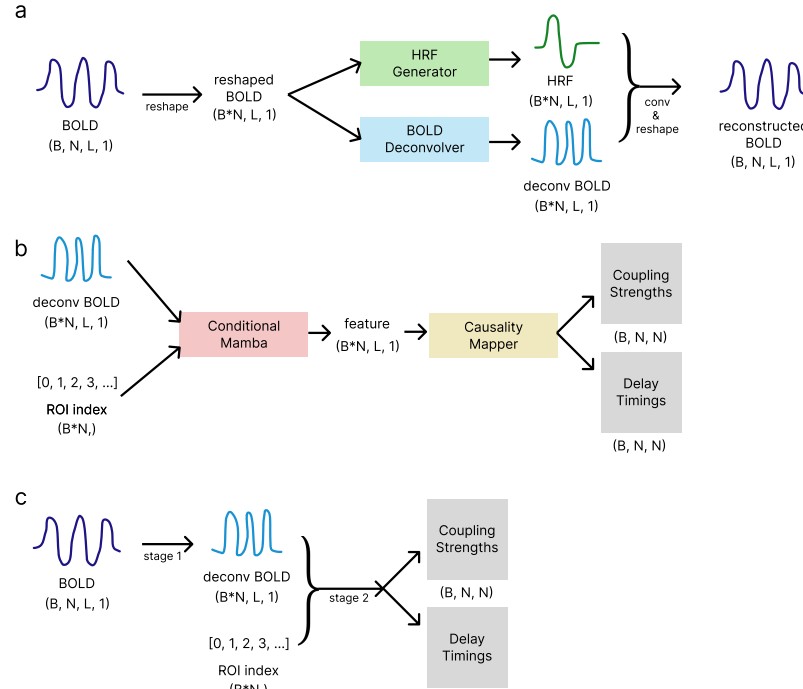

Figure 1: **Framework of CausalMamba.** **(a)** Stage 1: Hemodynamic Deconvolution. The model takes a BOLD signal and jointly learns a region-specific Hemodynamic Response Function (HRF) and its underlying latent neural activity. These two components are then convolved to reconstruct the original BOLD signal, guiding training. **(b)** Stage 2: Causal Inference. A Conditional Mamba encoder maps the deconvolved neural activity into feature representations. A causality mapper then uses these features to estimate a directed causal graph, defined by coupling strengths and delay timings between regions. ROI-index conditioning is controlled by ablation (shuffling, shared adapters) to mitigate memorization. **(c)** Integrated Pipeline. The full model is fine-tuned end-to-end using a three-stage curriculum. All training occurs on simulated data; real HCP data is used only for evaluation.

Our proposed model, **CausalMamba** (Figure 1), is designed to address the ill-posed inverse problem of inferring neural causality directly from BOLD signals. We reformulate this challenge by decomposing it into two more tractable sub-problems: (1) deconvolving the BOLD signal to estimate latent neural activity traces that approximate the underlying neuronal drivers and associated metabolic demand of the fMRI signal, and (2) mapping the inferred deconvolved BOLD activity to a directed causal graph. To effectively train this multi-part architecture, we employ a three-stage curriculum learning strategy.

### 3.2.1 Stage 1: BOLD Deconvolution

The objective of Stage 1 is to solve the **hemodynamic inverse problem** by mapping observed BOLD signals to latent neural activity through a **differentiable forward model** rather than assuming a fixed canonical kernel. This is achieved via two parallel modules operating on the input BOLD signal: an HRF Generator and a BOLD Deconvolver (Figure 2). To account for neurovascular heterogeneity across both regions and individuals (Handwerker et al. (2004); Taylor et al. (2018); Baron et al. (2025)), the **HRF Generator** uses a Conditional Mamba encoder to regress physiologically-meaningful parameters of a double gamma HRF model, defining a unique, subject- and ROI-specific Hemodynamic Response Function (HRF). Details are described in Appendix C. Concurrently, the **BOLD Deconvolver** infers latent LFP-like neural signals by using cross-attention between Conditional Mamba features and learnable queries to predict three key signal parameters: peak amplitude,

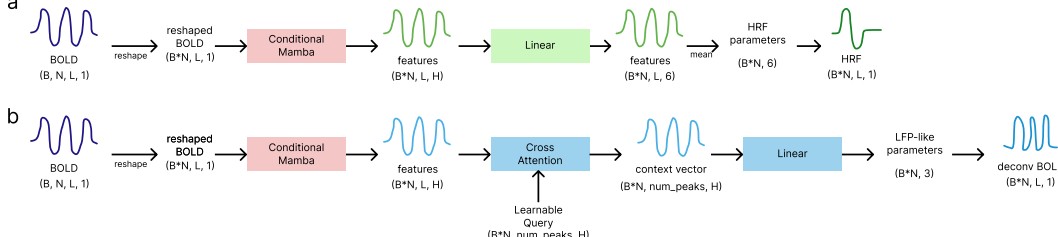

Figure 2: **Architecture of Stage 1 BOLD Deconvolution Modules.** The two parallel modules that map the observed BOLD signal to its underlying neural activity are illustrated. **(a)** The HRF Generator uses a Conditional Mamba encoder followed by a linear layer to regress the six parameters defining a unique, subject- and ROI-specific HRF.**(b)** Concurrently, the BOLD Deconvolver uses a Conditional Mamba encoder and a cross-attention mechanism with learnable queries to infer the parameters (amplitude, timing, width) of the latent LFP-like neural signal.

timing, and width. The differentiable forward process then reconstructs the BOLD signal by convolving the inferred neural signal with the generated HRF, enabling end-to-end gradient flow. To mitigate potential memorization of ROI identity from region-specific parameters, we additionally perform ablation experiments with shuffled ROI indices and group-shared adapters (see Appendix K for details). Stage 1 is trained with a composite objective (Appendix E, Equation 4) that combines an L1 BOLD reconstruction error ($L_{BOLD}$) with losses on predicted neural event parameters ($L_{amplitude}$, $L_{width}$). To ensure temporal fidelity, we introduce a composite timing loss ($L_{timing}$) that penalizes absolute, relative, and interval errors in event timings. For more details, see Appendix E.

### 3.2.2 STAGE 2: NEURAL ACTIVITY-TO-CAUSALITY MAPPING

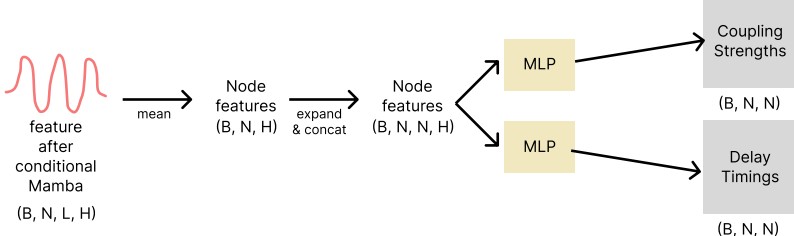

Figure 3: **Architecture of Stage 2 Causality Mapper.** This module infers the final causal graph from the feature representation generated by the Conditional Mamba backbone in Stage 2. To preserve temporal structure, the time-series features are first averaged to produce node-level feature vectors for each ROI. These vectors are then expanded and concatenated to form pairwise features, which are passed to two lightweight MLPs—rather than deeper graph modules—to directly predict the final Coupling Strengths and Delay Timings between all pairs of ROIs, thereby maintaining model simplicity and preserving temporal fidelity.

In Stage 2 (Figure 1 b), we train a network to infer the directed causal graph—defined by coupling strengths and delay timings—from the deconvolved BOLD signals estimated in Stage 1. This module consists of a Conditional Mamba backbone and a Causality Mapper. For the final causality mapping, we found that a direct projection from the backbone features was superior to using intermediate graph networks. Our Causality Mapper first creates a robust summary vector for each ROI by temporally averaging the sequence features from the Conditional Mamba backbone. These summary vectors are then expanded to form pairwise inputs for two lightweight MLPs that predict the final coupling strengths and delay timings (Figure 3). As demonstrated in our extensive ablation studies (Appendix K), introducing an additional GNN layer (e.g., GAT or GCN) between the backbone and this final mapper consistently degraded performance. We therefore hypothesize that the features learned by Conditional Mamba are already sufficiently rich for direct causal inference,

and that intermediate graph-based transformations risk over-processing and distorting this crucial information.

### 3.2.3 STAGE 3: END-TO-END FINE-TUNING OF THE COMPLETE PIPELINE

In Stage 3 (Figure 1 c), the full pipeline is fine-tuned end-to-end, initialized with the pre-trained weights from the previous stages. Note that all training in Stages 1–3 is performed exclusively on simulated datasets; real HCP task fMRI data are used for evaluation only. The complete model is then fine-tuned with a substantially lower learning rate of 1e-5, wo orders of magnitude smaller than the 1e-3 rate used in the preceding stages. This differential learning rate strategy stabilizes optimization of the composite system while retaining the specialized representations learned during earlier curriculum stages.

### 3.3 CONDITIONAL MAMBA

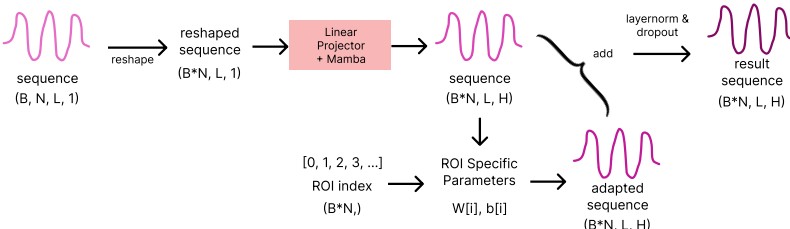

Figure 4: **Architecture of Conditional Mamba.** This module constitutes the central architectural element of our approach. The main Mamba pathway learns universal, ROI-agnostic temporal dynamics. In parallel, an adaptive stream uses the ROI index to select dedicated weights (W[i]) and biases (b[i]) that apply a region-specific affine modulation to the main pathway's output. The two streams are then integrated via a residual connection, enabling the model to combine global dynamics with region-specific characteristics while mitigating the risk of the model memorizing ROI identifiers rather than learning true neurovascular heterogeneity.

---

**Algorithm 1** Conditional Mamba Encoder

---

**Input:** Batch time series $\mathbf{X} \in \mathbb{R}^{(B \cdot N) \times L \times D_{\text{in}}}$, ROI indices $\text{roi\_ids} \in \{0, \ldots, N_{\text{ROI}} - 1\}^{(B \cdot N)}$
**Output:** ROI-conditioned rep. $\mathbf{Y} \in \mathbb{R}^{(B \cdot N) \times L \times D_{\text{hidden}}}$
  1:  ▷ Linear projection to hidden dimension
  2: $\mathbf{H}_{\text{proj}} \leftarrow \text{Linear}(\mathbf{X})$
  3:  ▷ ROI-agnostic temporal features via base Mamba encoder
  4: $\mathbf{H}_{\text{base}} \leftarrow \text{Mamba}(\mathbf{H}_{\text{proj}})$
  5:  ▷ Select ROI-specific adapter parameters (batched indexing)
  6: $\mathbf{W} \leftarrow \text{roi\_adapter\_weights}[\text{roi\_ids}]$  ▷ shape $(B \cdot N, D_h, D_h)$
  7: $\mathbf{b} \leftarrow \text{roi\_adapter\_bias}[\text{roi\_ids}]$  ▷ shape $(B \cdot N, D_h)$
  8:  ▷ Region-specific affine modulation
  9: $\mathbf{H}_{\text{adapt}} \leftarrow \text{BatchMatMul}(\mathbf{H}_{\text{base}}, \mathbf{W}) + \mathbf{b}\,\text{unsqueeze}(1)$
 10:  ▷ Residual connection and normalization
 11: $\mathbf{H}_{\text{out}} \leftarrow \mathbf{H}_{\text{base}} + \mathbf{H}_{\text{adapt}}$
 12: $\mathbf{Y} \leftarrow \text{Dropout}(\text{LayerNorm}(\mathbf{H}_{\text{out}}))$
 13: **return Y**

---

Conditional Mamba augments the vanilla Mamba SSM by introducing a parallel, adaptive stream with ROI-specific parameters. This stream uses dedicated, learnable weight matrices ($W[i]$) and biases ($b[i]$) for each ROI index, which allows for more complex and expressive transformations tailored to each ROI's unique characteristics, moving beyond simple scale-and-shift adjustments. These parameters are used to generate multiplicative modulations of the main pathway's output. The resulting transformed features are then integrated back into the main stream via a residual connection. This integration method is crucial for stable learning, as it allows local information to flexibly adjust the universal features without the risk of overwriting them.

This architectural design separates the learning roles of the two streams. The main Mamba pathway focuses on learning universal, ROI-agnostic temporal dynamics, while the lightweight adaptive stream specializes in region-specific modulations. This role separation stabilizes optimization and improves efficiency, enabling the model to tackle global and local patterns without conflating them.

Consequently, the Conditional Mamba structure is designed to capture the dual nature of brain data—universal temporal principles and region-specific dynamics—and thereby enables causal inference that is more accurate, stable, and neuroscientifically interpretable.

### 3.4 METRICS

We evaluate our models with metrics tailored for simulated and real-world data. For simulated data with known ground truth, we use **Coupling Loss** (L1 distance) to measure regression performance and **Causality Accuracy** for the ternary classification of connections (excitatory, inhibitory, or absent). For real HCP fMRI data, we report link-level **F1 scores** for classifying connections as positive, negative, or present. However, because link-level scores alone cannot capture the systems-level integrity of multi-step neural circuits, we introduce the **Known Pathway Recovery Rate (KPRR)**. KPRR is the proportion of subjects for whom a complete, literature-established pathway (e.g., V1→V2→V4 including self-inhibition) is correctly recovered, directly quantifying the model's ability to identify canonical circuits. For a fair comparison, all evaluation thresholds are fixed per method on a validation set (see Appendix D for details).

## 4 RESULTS

### 4.1 CAUSALITY INFERENCE IN SIMULATED DATA

| Model | Causality Accuracy | Coupling Loss | Jaccard Score |
|---|---|---|---|
| DCM | 0.4682 | 0.2934 | 0.2430 |
| rDCM | 0.4464 | 0.3002 | 0.1488 |
| Granger Causality | 0.4732 | – | 0.2147 |
| Neural Granger Causality | 0.4167 | – | 0.1611 |
| Mamba-backbone | 0.5466±0.0037 | 0.1633±0.0009 | 0.1892±0.0089 |
| GRU-backbone | 0.5431±0.0050 | 0.1663±0.0010 | 0.1965±0.0036 |
| conditional GRU-backbone | 0.6406±0.0027 | 0.1363±0.0001 | 0.2714±0.0012 |
| CausalMamba (ours) | **0.6496±0.0004** | **0.1347±0.0010** | **0.2754±0.0043** |

Table 1: **Performance comparison with baseline models on 6-ROI simulation data.** Coupling Loss was not computed for Granger Causality-based models (–) because they determine the presence or absence of a causal link (a binary output) rather than estimating its continuous strength.

On simulated 6-ROI data, **CausalMamba** significantly outperforms all baselines, achieving a 38.7% accuracy gain over DCM variants (e.g., rDCM, $p < 0.001$). This performance is driven by our novel ROI-conditioning strategy, which boosts accuracy by over 18% compared to a non-conditioned backbone ($p < 0.01$), and by the Mamba architecture's superiority over GRU. All other metrics, including Coupling Loss and Jaccard Score, also showed marked improvements over baselines (Table 1). Qualitatively, our model's estimates closely match the ground truth, whereas baselines exhibit known flaws; Granger Causality fails to capture continuous strengths, and DCM tends to underestimate interaction magnitudes (Figure 5). Beyond accuracy, CausalMamba offers superior computational scalability. Its cost scales linearly with the number of ROIs, in contrast to the quadratic scaling of DCM, making our model well-suited for large-scale network analysis (see Appendix, Figure S1).

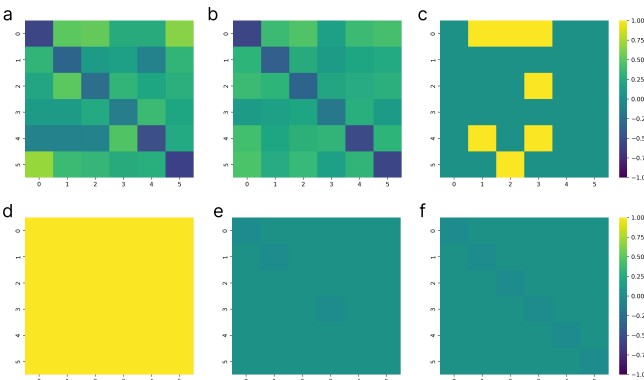

Figure 5: **Signed, Directed Coupling on the 6-ROI Simulation.** Each panel shows a signed, directed coupling matrix with *rows = sources (from)* and *columns = targets (to)*; colors encode normalized weights in $[-1, 1]$ with a zero-centered colormap identical across panels. Diagonal entries denote self-inhibition when present. **a** ground-truth coupling; **b** CausalMamba; **c** Granger Causality; **d** Neural Granger Causality; **e** rDCM; **f** DCM. All methods were evaluated under the same simulated data and preprocessing; additional metrics are summarized in the main results table.

| Strategy | Causality Accuracy | Coupling Loss |
|---|---|---|
| BOLD to Causality | 0.5934±0.0052 | 0.1512±0.0022 |
| CausalMamba, Only Stage 3 (end-to-end) | 0.6070±0.0179 | 0.1562±0.0009 |
| CausalMamba, Without Stage 2 Pre-training | 0.6207±0.0018 | 0.1487±0.0021 |
| CausalMamba, Without Stage 1 Pre-training | 0.6237±0.0085 | 0.1452±0.0024 |
| CausalMamba (ours) | **0.6496±0.0040** | **0.1347±0.0010** |

Table 2: Ablation study on problem decomposition and curriculum learning strategies.

Ablation studies support our staged, curriculum-based approach (Table 2). Omitting either the Stage 1 (BOLD deconvolution) or Stage 2 (causality mapping) pre-training degrades accuracy by 4.66% and 4.15%, respectively. This comparable degradation highlights that addressing hemodynamic confounds and learning the subsequent causal mapping are equally critical components. Furthermore, non-curriculum strategies like direct BOLD-to-causality training or end-to-end training from scratch result in far greater performance drops of 9.47% and 7.02%. This underscores the necessity of our curriculum strategy for achieving stable convergence on this challenging inverse problem.

## 4.2 CAUSALITY INFERENCE IN REAL DATA (HCP)

### 4.2.1 VALIDATION ON A CANONICAL PATHWAY

To validate our model on real data, we analyzed the canonical 3-ROI visual pathway (V1→V2→V4) from the HCP S1200 (n=1081) dataset. As summarized in Table 3, **CausalMamba** successfully recovered the full pathway in 88% of subjects (KPRR=0.88) with strong, balanced link-level performance. In stark contrast, all conventional methods failed at the systems level (KPRR < 0.01), unable to identify the canonical chain. These baselines also showed critical link-level flaws: rDCM exhibited a strong negative-edge bias, while GC-based models could not capture inhibitory connections at all. These failures stem from the baselines' methodological constraints (e.g., fixed hemodynamic assumptions), whereas our approach proved robust. Further analyses on a more complex pathway are provided in Appendix J.

| Model | KPRR | F1 pos. | F1 neg. | F1 presence | F1 macro |
|---|---|---|---|---|---|
| DCM | 0.0000 | 0.6760 | 0.6308 | 0.6684 | 0.6534 |
| rDCM | 0.0074 | 0.2638 | 0.9431 | 0.7657 | 0.6034 |
| Granger Causality | 0.0000 | 0.4615 | 0.0000 | 0.2621 | 0.2308 |
| Neural Granger Causality | 0.0000 | 0.5000 | 0.0000 | 0.9090 | 0.2499 |
| **CausalMamba (ours)** | **0.8807** | **0.7714** | **0.8009** | **0.7878** | **0.7862** |

Table 3: **Performance comparison with baseline models on 3-ROI HCP data.** KPRR denotes for the Known Pathway Recovery Rate (KPRR).

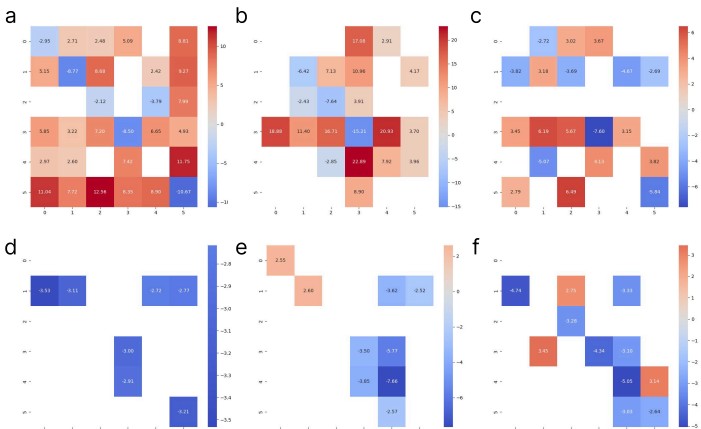

Figure 6: **Directed Coupling in a Working-Memory Task (2-back vs 0-back) on the HCP 6-ROI Network**. Panels show paired $t$-statistics for the coupling change ($\Delta w = w_{\text{2-back}} - w_{\text{0-back}}$) between high and low working-memory loads. Rows indicate sources and columns are targets; red denotes strengthening and blue denotes weakening of coupling under high load. Only significant edges (Benjamini–Hochberg FDR corrected, $q < 0.05$) are shown. Diagonal entries denote self-inhibition. ROI indices: 0 = DLPFC, 1 = PPC, 2 = dACC, 3 = Left Insula, 4 = Right Insula, 5 = preSMA. **a–c**: CausalMamba; **d–f**: rDCM. Stimuli: **a,d** = body, **b,e** = faces, **c,f** = tools. Both methods used identical preprocessing.

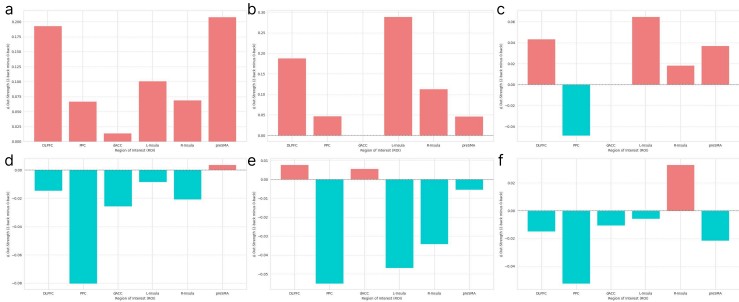

Figure 7: **Directed Coupling in a Working-Memory Task (2-back vs 0-back): ROI-wise $\Delta$ Out-Strength.** Bars show the change in outgoing causal influence per ROI, $\Delta s_i^{\text{out}} = \sum_j w_{i \to j}^{(2\text{-}back)} - \sum_j w_{i \to j}^{(0\text{-}back)}$, computed from signed, directed weights (higher vs lower working-memory load). Positive bars indicate higher outgoing influence under 2-back; negative bars indicate reductions. **a–c**: CausalMamba; **d–f**: rDCM. Stimuli: **a,d** = body; **b,e** = faces; **c,f** = tools. Across stimuli, CausalMamba shows stimulus-contingent increases at distinct ROIs (e.g., L-Insula for faces; preSMA for body), whereas rDCM estimates predominantly decreases across ROIs. All methods use identical preprocessing and the same atlas; ROI labels on the $x$-axis are DLPFC, PPC, dACC, L-Insula, R-Insula, and preSMA.

### 4.2.2 WORKING-MEMORY (N-BACK): STIMULUS-SPECIFIC HUB SHIFTS IN DIRECTED CONNECTIVITY

To see if CausalMamba could detect how the brain flexibly reorganizes itself, we used data from a working memory experiment. This task contrasted a low-load attention condition (0-back) with a high-load working memory condition requiring recall of the image from two trials prior (2-back). CausalMamba revealed that the brain's primary driver regions—or causal hubs—strategically shifted depending on both the mental workload and the type of image being viewed. For example, during the high-effort task, seeing "Faces" increased the causal influence from the Left Insula, whereas seeing "Body" images increased influence from the DLPFC and preSMA regions (Figure 6, 7). In stark contrast, a conventional method (rDCM) simply estimated that brain connectivity uniformly decreased. CausalMamba thus uncovers the precise, task-dependent network reorganizations that traditional methods miss.

### 4.3 ANALYSIS OF MODEL ROBUSTNESS TO INCREASED ROI COUNTS

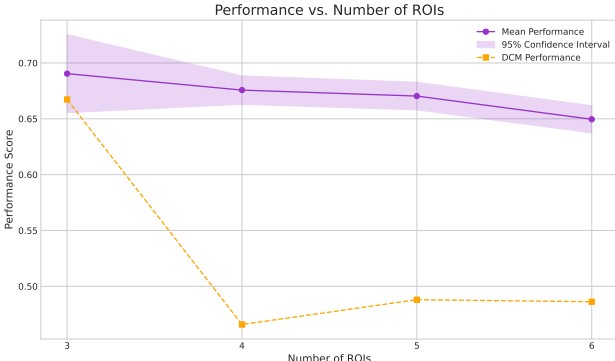

Figure 8: **Performance comparison with respect to the number of ROIs.** The plot illustrates the performance scalability of our proposed model (solid purple line) against the DCM baseline (dashed orange line) as the number of ROIs increases from 3 to 6. The shaded area represents the 95% confidence interval for our model's performance across three independent runs.

We assessed model scalability against DCM by increasing network complexity from 3 to 6 ROIs. Our model demonstrated high stability, with performance decreasing only slightly from 0.69 to 0.65. In stark contrast, DCM, while comparable at 3 ROIs (0.667), suffered a severe performance drop at 4 ROIs (0.466). This clearly indicates that CausalMamba can effectively scale to more complex systems where traditional methods like DCM become unreliable.

## 5 CONCLUSION

We introduced **CausalMamba**, a region-adaptive state-space framework for fMRI causal inference that tackles ill-posedness via a two-stage pipeline—BOLD deconvolution followed by ROI-conditioned causality mapping. The model consistently surpassed representative DCM variants and other baselines in simulations and remained robust to HRF and SNR perturbations. On HCP data, CausalMamba recovered the canonical V1→V2→V4 pathway with high system-level fidelity, while conventional methods showed near-zero KPRR. It further captured load- and stimulus-dependent hub shifts in the n-back contrast (e.g., DLPFC for Faces, Right Insula for Body/Tools), revealing condition-dependent reconfiguration of directed connectivity that standard approaches often miss. The framework scales approximately linearly with ROI count ($\mathcal{O}(N)$) for practical large-network analyses. Current limitations include block-level contrasts that do not isolate event-locked processes, partial control of HRF variability, and a uni-directional, single-lag formulation. Future work will model HRF uncertainty, extend to bi-directional multi-lag dynamics with calibrated confidence intervals, evaluate cross-task generalization, and release code, weights, and simulators for reproducibility.

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

# A  SIMULATED DATA GENERATION

To train and evaluate our model, we synthesized a large-scale dataset comprising 10,000 unique samples. Each sample simulates brain activity from a four-node network for 300 seconds with a sampling resolution (dt) of 0.1 seconds. We downsampled with a sampling resolution of 0.8 seconds for a realistic data setting. The generation process is designed to produce biophysically plausible data by integrating realistic neural dynamics with heterogeneous hemodynamic responses.

## A.1  NEURAL MASS MODEL AND CONNECTIVITY

The core of our simulation is a $N$-node neural mass model, where each node represents a distinct Region of Interest (ROI). The interactions between these nodes are defined by a $N \times N$ connectivity matrix, encompassing both connection strength and signal transmission delay.

- Connectivity Strengths: The coupling parameters were stochastically generated to reflect biological principles. Self-connections (diagonal elements) were predominantly inhibitory, representing local feedback inhibition (Youssofzadeh et al. (2015)). Each node had a 75% probability of featuring a self-inhibitory connection, with its strength sampled from a uniform distribution. The specific ranges were tailored to mimic different cortical regions: strong inhibition for a putative frontal ROI (e.g., [-0.3, -0.7]), moderate for parietal and temporal ROIs, and weak for an occipital ROI (Zavaglia et al. (2010)). Self-excitation was excluded as it is biologically rare. Inter-regional connections (off-diagonal elements) were primarily excitatory (90% probability, strength uniformly sampled from [0.2, 0.8]), with a 10% chance of being inhibitory ([-0.1, -0.3]) (Tjia et al. (2017); Sukenik et al. (2021)).
- Transmission Delays: Time delays for inter-regional connections were sampled from a uniform distribution between 0.5 and 3.0 seconds, while delays for self-inhibitory connections were shorter, ranging from 0.2 to 1.2 seconds (Ton et al. (2014)).

## A.2 Neural Activity Simulation

Our simulation aims to generate neural timeseries analogous to LFP, representing the summed, dynamic activity of a neuronal population. For each sample, we first simulated the intrinsic neural activity for every ROI before applying coupling effects.

- Exogenous Input: An external stimulus was administered exclusively to the first ROI. This stimulus consisted of 12 discrete events of 2-second duration, occurring at semi-random intervals. The resulting boxcar function was smoothed with a Gaussian kernel ($\sigma = 3.0$) and passed through a hyperbolic tangent activation function to create a more realistic, smooth input signal (Deco et al. (2011); Breakspear (2017)).

- Intrinsic Dynamics: The base neural activity for each ROI was a composite signal, summing the external input (for ROI 1 only), spontaneous physiological fluctuations modeled as smoothed Gaussian noise, and scale-free pink noise. This composite input was then transformed using a rectified linear unit (ReLU) followed by a gain factor and baseline addition, emulating the non-negative and dynamic nature of neuronal firing rates (Deco et al. (2009); Bedard et al. (2006)).

- Coupled Activity: The final neural activity for each node was calculated by adding the delayed and weighted contributions from all other nodes (as defined by the coupling matrices) to its intrinsic activity (Jansen & Rit (1995); David & Friston (2003); Spiegler et al. (2011)).

## A.3 Hemodynamic Forward Model and BOLD Signal Generation

The simulated neural activity was translated into a Blood-Oxygen-Level-Dependent (BOLD) fMRI signal using a forward hemodynamic model.

- Hemodynamic Response Function (HRF): To account for regional neurovascular heterogeneity, each of the four ROIs was assigned a unique HRF. The HRF was modeled using a double-gamma function (Friston et al. (1998)). Its key parameters—such as 'peak delay' (4.0-9.5s), 'undershoot delay' (12.0-24.0s), and 'undershoot scale'—were not fixed but were stochastically drawn from distributions whose means and variances were chosen to be characteristic of different major cortical lobes (e.g., faster responses for frontal, slower for temporal) (Handwerker et al. (2004); Aguirre et al. (1998)).

- Convolution and Scaling: The final neural activity of each ROI was convolved with its corresponding unique HRF to produce a raw BOLD signal. This signal was then normalized to a target Percent Signal Change (PSC), with the peak amplitude randomly set between 0.8% and 2.5%, and scaled to a baseline mean value of 100 (Glover (1999); Lindquist et al. (2009)).

## A.4 Feature Extraction and Final Dataset

As a final preprocessing step, we extracted key features from the simulated neural time series to serve as ground truth for model inversion.

- Neural Event Detection: We applied a peak-finding algorithm ($scipy.signal.find_peaks$) to the neural activity of each ROI to identify the timing, amplitude, and width of significant neural events (Quiroga et al. (2004); Maccione et al. (2009)).

- Standardization: Since the number of detected events varied per simulation, the resulting feature vectors (timings, amplitudes, widths) were padded with zeros or truncated to a fixed length of 12 events per ROI, corresponding to the number of external stimuli (Faust et al. (2018)).

The final saved dataset for each of the 10,000 samples includes the ground-truth generating parameters (connectivity and HRF), the full time series for both neural activity and the BOLD signal for all four ROIs, and the extracted, standardized neural event features.

## B  HCP TASK FMRI DATA PREPROCESSING

Following prior rDCM studies Frässle et al. (2021), we adopted a motor task network comprising visual, supplementary motor area (SMA), and primary motor cortex (M1) regions, leveraging the well-established visuomotor pathway. We utilized preprocessed HCP S1200 motor task fMRI data (LR run) and applied the HCPMMP1 atlas to extract region-of-interest (ROI) time series. Specifically, we selected five key regions: V1, V2, V4, SMA, and M1 for our causal analysis framework. Since known pathways cannot be guaranteed as ground truth, we trained our model on simulated data with verifiable ground truth and performed inference only on HCP data for validation.

## C  HRF PARAMETERS

The HRF Generator regresses six physiologically-meaningful parameters of a double gamma HRF model:

$$h(t) = A \cdot \left(\frac{t}{t_p}\right)^{\alpha_1} e^{-\frac{t-t_p}{\beta_1}} - a_u \cdot \left(\frac{t}{t_u}\right)^{\alpha_2} e^{-\frac{t-t_u}{\beta_2}} \tag{1}$$

where the six learnable parameters are:

- $t_p$ (peak time): Time to peak response (range: 4-8s, default: 6s)
- $t_u$ (undershoot time): Time to undershoot (range: 12-20s, default: 16s)
- $A$ (peak amplitude): Maximum signal change (range: 0.5-2.0)
- $a_u$ (undershoot ratio): Undershoot amplitude ratio (range: 0.1-0.5)
- $\alpha_1, \alpha_2$ (shape parameters): Control rise time (range: 5-10)

These parameters capture ROI-specific neurovascular coupling variations, as different brain regions exhibit distinct hemodynamic properties (Handwerker et al. (2004)).

## D  METRICS

### D.1  CAUSALITY ACCURACY

To evaluate our model's performance, we quantify its ability to accurately recover the ground-truth causal connectivity graph from the input BOLD signals. This evaluation involves a two-step process: (1) post-processing the model's continuous-valued output into a discrete, ternary connectivity matrix, and (2) calculating the element-wise accuracy of this matrix against the ground-truth matrix.

First, the model's raw output, a matrix of continuous coupling strengths $\mathbf{S} \in \mathbb{R}^{N \times N}$, is converted into a ternary adjacency matrix $\hat{\mathbf{C}} \in \{-1, 0, 1\}^{N \times N}$. This matrix represents directed causal links, where '+1' denotes an excitatory link, '-1' denotes an inhibitory link, and '0' indicates no significant connection. This discretization is performed based on biologically-informed rules:

**Self-Connections (Diagonal Elements)**  The diagonal elements of $\mathbf{S}$ represent self-regulation within a Region of Interest (ROI). These are categorized using a predefined threshold, $\tau_{\text{self}}$. $\tau_{\text{self}}$ is empirically set as 0.1, based on previous studies (David & Friston (2003); Jansen & Rit (1995)). This value was empirically determined to be optimal through a sensitivity analysis detailed in Appendix I. A diagonal value $S_{ii} > \tau_{\text{self}}$ is classified as self-excitation (+1), while $S_{ii} < -\tau_{\text{self}}$ is classified as self-inhibition (-1). Values where $|S_{ii}| < \tau_{\text{self}}$ are considered insignificant and set to 0. This approach aligns with our simulation design, which primarily features self-inhibitory connections while excluding self-excitation.

**Inter-regional Connections (Off-Diagonal Elements)**  To resolve the direction of influence between any two regions and establish directed connections, we employ an asymmetry principle. A directed causal link from region $i$ to region $j$ is established ($\hat{C}_{ij} = 1$) if and only if the corresponding coupling strength is greater than the strength in the reverse direction ($S_{ij} > S_{ji}$). In this case, the

reverse connection is set to zero ($\hat{C}_{ji} = 0$) to enforce a single dominant direction. If the strengths are equal, or if neither dominates, both are set to zero.

Once the predicted matrix $\hat{\mathbf{C}}$ is generated, we compute the **Causality Accuracy** by comparing it element-wise to the ground-truth matrix $\mathbf{C}$. The accuracy is defined as the fraction of correctly classified elements (including the diagonal) across the entire matrix, averaged over all samples in a batch. Formally, for a batch of $B$ samples, the accuracy is:

$$\text{Accuracy} = \frac{1}{B} \sum_{b=1}^{B} \frac{1}{N^2} \sum_{i=1}^{N} \sum_{j=1}^{N} \mathbb{I}(\hat{C}_{ij}^{(b)} = C_{ij}^{(b)}) \tag{2}$$

where $\mathbb{I}(\cdot)$ is the indicator function, $N$ is the number of ROIs, and $\hat{\mathbf{C}}^{(b)}$ and $\mathbf{C}^{(b)}$ are the predicted and true matrices for the $b$-th sample in the batch, respectively.

### D.2 KNOWN PATHWAY RECOVERY RATE

To evaluate model performance on the real-world HCP task-fMRI dataset, where a complete ground-truth connectivity matrix is unavailable, we assess each model's ability to identify well-established neuroscientific pathways. We define a metric, the **Known Pathway Recovery Rate**, which measures the proportion of subjects for whom a predefined canonical pathway is successfully identified.

For our analyses, we targeted two distinct pathways based on prior literature:

- **3-ROI Visual Pathway**: A segment of the canonical visual stream, defined as a directed chain of connections from V1 to V2, and then to V4 (i.e., $V1 \rightarrow V2 \rightarrow V4$), including self-regulatory pathways.

- **4-ROI Visuomotor Pathway**: A functional pathway relevant to the motor task, defined as $V1 \rightarrow V2 \rightarrow \text{SMA} \rightarrow M1$, including self-regulatory pathways.

The calculation process is as follows. For each of the 1,081 subjects, we first convert the model's raw output of coupling strengths into a discrete ternary connectivity matrix, $\hat{\mathbf{C}}$, as described previously. We then check if all constituent links of the target known pathway are present in this matrix (i.e., the corresponding elements have a value of $+1$). A subject is assigned a score of 1 if the entire pathway is recovered, and 0 otherwise. The final Known Pathway Recovery Rate is the average score across all subjects in the cohort. Formally:

$$\text{Recovery Rate} = \frac{1}{N_{\text{subjects}}} \sum_{s=1}^{N_{\text{subjects}}} \mathbb{I}(\mathcal{P} \subseteq \hat{\mathbf{C}}^{(s)}) \tag{3}$$

where $N_{\text{subjects}}$ is the total number of subjects (1081), $\mathcal{P}$ is the set of directed links comprising the known pathway, and $\mathbb{I}(\mathcal{P} \subseteq \hat{\mathbf{C}}^{(s)})$ is an indicator function that is 1 if all links in $\mathcal{P}$ are present in the predicted matrix for subject $s$, and 0 otherwise.

### D.3 LINK-LEVEL F1 METRICS (POSITIVE/NEGATIVE/PRESENCE)

**Setup.** Let $S \in \mathbb{R}^{n \times n}$ denote the predicted coupling matrix where larger values indicate stronger $i \rightarrow j$ influence, and $Y \in \{-1, 0, +1\}^{n \times n}$ the ground-truth labels ($-1$: inhibitory, 0: no edge, $+1$: excitatory). We evaluate edges over a candidate set $\Omega \subseteq \{(i, j) \mid i \neq j\}$. Unless stated otherwise, we use a *within-pathway* $\Omega$ (restricted to the pathway nodes) and also report a *global* variant as a robustness check. Self-edges are treated as negative ($-1$) by default; results with self-edges excluded are provided for completeness.

**Dominant-direction rule.** To avoid double-counting contradictory directions, we optionally apply a *dominant* rule: for each unordered pair $\{i, j\}$ we keep only the direction with the larger absolute score, $\max\{|S_{ij}|, |S_{ji}|\}$, removing the other from $\Omega$ (self-edges unaffected). We report with/without this rule.

**Thresholded prediction.** We form sign predictions at thresholds $\tau_+, \tau_- > 0$ via

$$\hat{y}_{ij}(\tau_+, \tau_-) = \begin{cases} +1 & \text{if } S_{ij} \geq \tau_+ \\ -1 & \text{if } S_{ij} \leq -\tau_- \quad (i,j) \in \Omega. \\ 0 & \text{otherwise} \end{cases}$$

Presence (edge vs. no-edge, ignoring sign) is predicted by

$$\hat{z}_{ij}(\tau_{\text{abs}}) = \mathbf{1}\big[\, |S_{ij}| \geq \tau_{\text{abs}} \,\big], \qquad \tau_{\text{abs}} = \min(\tau_+, \tau_-).$$

**Precision/Recall/F1.** For each binary task—**positive** ($+1$ vs. rest), **negative** ($-1$ vs. rest), and **presence** ($\pm$ vs. 0)—we compute

$$\text{Precision} = \frac{\text{TP}}{\text{TP} + \text{FP}}, \quad \text{Recall} = \frac{\text{TP}}{\text{TP} + \text{FN}}, \quad \text{F1} = \frac{2 \cdot \text{Precision} \cdot \text{Recall}}{\text{Precision} + \text{Recall}}.$$

We report $\text{F1}_{\text{pos}}$, $\text{F1}_{\text{neg}}$, and $\text{F1}_{\text{presence}}$. Our macro sign score averages the two signed classes:

$$\text{F1}_{\text{macro-sign}} = \tfrac{1}{2}\big(\text{F1}_{\text{pos}} + \text{F1}_{\text{neg}}\big).$$

**Threshold selection and fairness.** Because different methods produce scores on different scales (e.g., DCM often yields $\mathcal{O}(10^{-3} \sim 10^{-2})$), we *fix* $(\tau_+, \tau_-)$ *per method* on a validation set and apply them unchanged to the test set:

$$(\tau_+^*, \tau_-^*) = \arg\max_{\tau_+, \tau_-} \ \tfrac{1}{2}\,\text{F1}_{\text{macro-sign}} + \tfrac{1}{2}\,\text{F1}_{\text{presence}}.$$

As a scale-invariant robustness check, we also provide a normalized variant $\tilde{S} = S / \big(\text{quantile}(|S|, 0.99) + \varepsilon\big)$ and repeat the evaluation with a *shared* $(\bar{\tau}_+, \bar{\tau}_-)$.

**Implementation notes.** We use the convention $S_{ij} \equiv i \to j$. For baselines whose adjacency uses $A_{ij} \equiv j \to i$, we transpose to align directions before evaluation.

# E  LOSS IMPLEMENTATION DETAILS

$$\begin{aligned}
L_{\text{stage3}} &= L_{\text{stage1}} + L_{\text{stage2}} \\
L_{\text{stage2}} &= L_1(coupling_{\text{true}}, coupling_{\text{pred}}) + L_1(delay_{\text{true}}, delay_{\text{pred}}) \\
L_{\text{stage1}} &= L_{\text{BOLD}} + 0.3 \cdot L_{\text{timing}} + L_{\text{width}} + L_{\text{amplitude}} \\
\text{where} \quad L_{\text{timing}} &= L_1(t_{\text{true}}, t_{\text{pred}}) + L_{\text{center\_shift}} + L_{\text{interval}} \\
L_{\text{center\_shift}} &= \frac{1}{L}\sum_{i=1}^{L}\big((t_{\text{pred},i} - \bar{t}_{\text{pred}}) - (t_{\text{true},i} - \bar{t}_{\text{true}})\big)^2 \\
L_{\text{interval}} &= \frac{1}{L-1}\sum_{i=2}^{L}\big((t_{\text{pred},i} - t_{\text{pred},i-1}) - (t_{\text{true},i} - t_{\text{true},i-1})\big)^2 \\
L_{\text{BOLD}} &= L_1(\text{BOLD}_{\text{true}}, \text{BOLD}_{\text{recon}}) \\
L_{\text{width}} &= L_1(\text{width}_{\text{true}}, \text{width}_{\text{pred}}) \\
L_{\text{amplitude}} &= L_1(\text{mean}(\text{amp}_{\text{true}}), \text{mean}(\text{amp}_{\text{pred}}))
\end{aligned} \tag{4}$$

Our training objective enforces both biophysical validity and temporal precision. The **BOLD Reconstruction Loss** $L_{\text{BOLD}}$ minimizes the L1 error between ground-truth BOLD signals and those reconstructed from predicted neural events convolved with the generated HRF, thereby ensuring physiologically consistent deconvolution. **LFP Parameter Losses** ($L_{\text{width}}, L_{\text{amplitude}}$) supervise predicted LFP-like event parameters against ground-truth parameters. To refine temporal fidelity, the composite **Timing Loss** $L_{\text{timing}}$ (weight 0.3) integrates three complementary terms: (i) an absolute L1 error on event timings for *global accuracy*, (ii) an **Center Shift Loss** $L_{\text{center\_shift}}$—MSE between mean-centered timing vectors to capture *relative temporal alignment* across events, and (iii) an **Interval Consistency Loss** $L_{\text{interval}}$—MSE between inter-event intervals to enforce *cadence consistency*. This decomposition encourages the model to capture both absolute and relative temporal structures in neural activity while maintaining physiological plausibility.

# F  IMPLEMENTATION DETAILS

## F.1  PREPROCESSING PIPELINE

We used the minimally preprocessed HCP 1200 Subject Release (S1200, 2 mm isotropic), which includes gradient distortion correction, motion correction, EPI distortion correction, slice-timing correction, and normalization to MNI152 space, as described in Glasser et al. (2013). No additional smoothing or filtering was applied beyond the HCP default.

## F.2  TRAINING HYPERPARAMETERS

```
config = {
    'backbone': 'mamba',
    'mamba': {
        'd_model': 64,
        'd_state': 16,
        'd_conv': 4,
        'expand': 2,
    },
    'learning_rate_stage1': 1e-3,
    'learning_rate_stage2': 1e-3,
    'learning_rate_stage3': 1e-5,

    'batch_size': 128,
    'epochs': 100,
    'optimizer': 'AdamW',
    'weight_decay': 1e-2,
    'scheduler': 'Warmup+CosineAnnealingLR',
    'warmup_epochs': 5,
    'gradient_clip': 1.0,
    'hidden_size': 64,
}
```

## F.3  SIMULATION DATA GENERATION

Data generation process following two algorithms (Algorithm S1, S2).

---

**Algorithm S1** Synthetic fMRI Data Generation (HRF, 1/f noise, coupling, PSC)

---

**Input:** Number of samples $S$, time grid $t = 0 : \Delta t : T$, number of ROIs $R = 3$
**Input:** Functions: $\mathrm{CreateHRF}(\cdot)$ (double-gamma), $\mathrm{PinkNoise}(\cdot)$ $(1/f)$, $\mathrm{ReLU}(\cdot)$, $\mathrm{PSC}(\cdot)$ (percent-signal-change scaling)

1: **for** $s = 1, \ldots, S$ **do**
2:     Draw HRF parameters for each ROI: $\theta_i \sim \mathcal{N}(\mu, \Sigma)$ for $i = 1, \ldots, R$
3:     Build stimulus train with 12 onsets, smooth (Gaussian), then apply nonlinearity: $u(t) \leftarrow \tanh(2\,\mathrm{smooth}(u_0(t))) \cdot 0.8$
4:     Generate base neural activities for each ROI:
        For ROI $i$: spontaneous $\sim \mathcal{N}(0, \sigma_i)$ smoothed, add $\mathrm{PinkNoise}$; only $\mathrm{ROI}_1$ receives $u(t)$
        Apply rate nonlinearity: $n_i(t) \leftarrow b_i + g_i \cdot \mathrm{ReLU}(\text{intrinsic input})$
5:     Sample coupling strength matrix $W \in \mathbb{R}^{R \times R}$ and delay matrix $D \in \mathbb{R}_+^{R \times R}$; set integer delays $\Delta_{ij} \leftarrow D_{ij}/\Delta t$
6:     For each ROI $i$: compute coupled neural activity
        $x_i(t) \leftarrow n_i(t) + \sum_{j \neq i} W_{ij}\, n_j(t - \Delta_{ij})$; set initial segment to $n_i(t)$ to avoid invalid shifts
7:     Convolve with ROI-specific HRF: $y_i(t) \leftarrow (x_i * h(\theta_i))(t) \cdot \Delta t$
8:     Scale to percent-signal-change: $\tilde{y}_i(t) \leftarrow \mathrm{PSC}(y_i(t); \text{target peak} \in [0.8, 2.5]\%)$
9:     Convert to BOLD baseline units: $b_i(t) \leftarrow 100 \cdot \big(1 + \tilde{y}_i(t)/100\big)$
10:     Save tensors for sample $s$: neural activities $\{x_i(t)\}_{i=1}^R$ and BOLD $\{b_i(t)\}_{i=1}^R$
11: **return** dataset $\{(b(t), x(t), W, D, \theta)\}_{s=1}^S$

---

---

**Algorithm S2** Spike-Property Extraction and Preprocessing

---

**Input:** Saved neural activities $\{x_i(t)\}$ for each sample, time step $\Delta t$

1: **for** each sample $s$ **do**
2:     **for** $i = 1, \ldots, R$ **do**
3:         Detect peaks on $x_i(t)$ using prominence $= 0.5 \cdot \text{std}(x_i)$ and minimum distance $= 5\,\text{s}/\Delta t$
4:         Compute peak timing (s), amplitude, and half-maximum width for each detected peak
5:         Keep at most 12 peaks; if fewer than 12, zero-pad symmetrically to length 12
6:     Stack per-ROI arrays to shapes $(R, 12)$ for timings, amplitudes, widths
7:     Save preprocessed tensors for sample $s$
8: **return** preprocessed dataset with spike properties

---

## G    COMPUTATIONAL SCALABILITY ANALYSIS

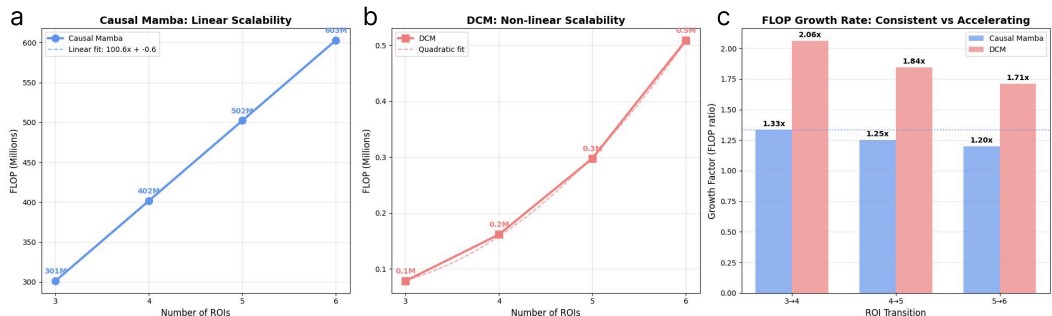

Figure S1:    **Computational Scalability Analysis of CausalMamba versus DCM.** The computational cost, measured in hardware-agnostic Floating-Point Operations (FLOPs), was evaluated as a function of the number of Regions of Interest (ROIs). **(a)** CausalMamba exhibits a **clear linear relationship** between the number of ROIs and the required FLOPs, as confirmed by a tight linear regression fit ($R^2 > 0.99$). **(b)** In stark contrast, DCM's computational cost shows **accelerating, non-linear growth**, which is well-approximated by a quadratic fit ($R^2 > 0.99$). Note the different y-axis scale, highlighting DCM's rapid increase in computational demand. **(c)** The FLOP growth factor between successive ROI counts reveals different scaling behaviors. CausalMamba maintains a **near-constant growth factor** ($\sim$1.25x), indicative of linear scaling, while DCM's growth factor is **significantly larger and accelerates** (from 2.06x to 1.71x), confirming its higher-order computational complexity.

To validate the scalability of our proposed framework, we conducted a computational cost analysis of CausalMamba against DCM. We measured the total number of **Floating-Point Operations (FLOPs)** required for a single inference pass as the number of network nodes (ROIs) was varied from 3 to 6. FLOPs serve as a standardized, hardware-agnostic metric for computational complexity, allowing for a fair comparison between models running on different architectures (e.g., GPU for CausalMamba, CPU for DCM).

The results, presented in **Figure S1**, unequivocally demonstrate the superior scalability of our method. CausalMamba's computational cost grows **linearly ($O(N)$)** with the number of ROIs (N), as shown by the excellent linear fit in **Figure S1a**. This predictable, efficient scaling is a core advantage of the underlying State-Space Model architecture.

Conversely, DCM exhibits a **non-linear, quadratic ($O(N^2)$) scaling** behavior (**Figure S1b**). The computational demands accelerate rapidly as more ROIs are added to the model. This is further clarified in **Figure S1c**, which shows that the factor of increase in FLOPs for each additional ROI is consistently low and stable for CausalMamba, whereas for DCM it is substantially higher and variable.

This fundamental difference in computational complexity confirms that CausalMamba is a practically scalable solution. Its linear scaling makes the causal analysis of large-scale, whole-brain networks (e.g., $> 100$ ROIs) computationally feasible. Indeed, we have confirmed in internal tests that our framework is computationally scalable to whole-brain analysis with 360 ROIs within a computationally feasible timeframe. This stands in stark contrast to the quadratic complexity of traditional methods like DCM, which becomes a prohibitive bottleneck for such large-scale analyses.

## H   MODEL SIZE

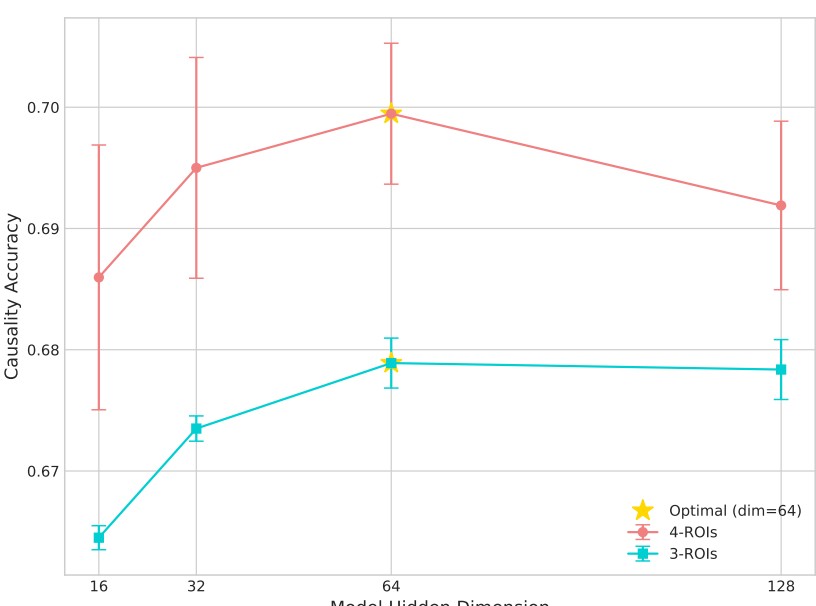

Figure S2: **Hyperparameter analysis for model hidden dimension.** The plot shows the Causality Accuracy as a function of the model's hidden dimension, evaluated on both 3-ROI (cyan squares) and 4-ROI (pink circles) simulation data. Each point represents the mean accuracy, with error bars indicating the standard deviation across multiple runs. Performance peaks at a hidden dimension of 64 for both scenarios, which is selected as the optimal hyperparameter (yellow star). This analysis reveals a clear trade-off, where dimensions smaller than 64 may lead to underfitting, while larger dimensions begin to show performance degradation, likely due to overfitting.

We conducted a comprehensive hyperparameter analysis to determine the optimal hidden dimension for the Conditional Mamba model. This process involved systematically evaluating the model's Causality Accuracy across a range of capacities—specifically, with hidden dimensions of 16, 32, 64, and 128—on both 3-ROI and 4-ROI simulation datasets. The goal was to identify the "sweet spot" where the model is complex enough to capture the underlying causal dynamics without becoming overly complex and losing its ability to generalize.

The results, visualized in Figure S2, demonstrate a clear and consistent trend. At smaller hidden dimensions like 16 and 32, the model likely has insufficient capacity to fully learn the intricate relationships within the data, a condition known as underfitting. As the model capacity increases from 16 to 64, we observe a significant improvement in performance for both experimental setups.

The performance peaks at a hidden dimension of 64. This is identified as the optimal dimension, where the model achieves the best balance between representational power and generalization. However, when the capacity is further increased to a hidden dimension of 128, the performance begins to degrade, particularly in the more complex 4-ROI scenario. This decline suggests that the model has become too powerful, leading to overfitting. An over-parameterized model starts to memorize noise

and specific artifacts of the training data rather than learning the true underlying patterns, which ultimately harms its accuracy on unseen test data.

Therefore, based on this empirical evidence, a hidden dimension of 64 was selected for our final model architecture to ensure maximal performance and robust generalization.

## I  HYPERPARAMETER SENSITIVITY FOR $\tau_{\text{SELF}}$

| $\tau_{\text{self}}$ | Causality Accuracy | Coupling Loss |
|---|---|---|
| 0.01 | 0.6462±0.0034 | 0.1354±0.0009 |
| 0.05 | 0.6476±0.0041 | 0.1353±0.0007 |
| **0.1** | **0.6496±0.0040** | **0.1347±0.0010** |
| 0.2 | 0.6418±0.0030 | 0.1352±0.0005 |

Table S1: Sensitivity analysis for the self-connection threshold ($\tau_{\text{self}}$). Results are reported as mean ± standard deviation over 3 random seeds. The best performance is highlighted in bold.

To validate the choice of the self-connection threshold, $\tau_{\text{self}}$, we performed a sensitivity analysis by evaluating the model's performance on a validation set with different threshold values: 0.01, 0.05, 0.1, 0.2. As shown in Table I, the model achieved the highest Causality Accuracy at $\tau_{\text{self}}=0.1$. Performance degraded for both lower and higher threshold values, confirming that 0.1 is a robust choice for this parameter.

## J  EXTENSION TO A MORE HETEROGENEOUS 4-ROI VISUOMOTOR PATHWAY

| Model | KPRR | F1 pos. | F1 neg. | F1 presence | F1 macro |
|---|---|---|---|---|---|
| DCM | 0.0000 | 0.4561 | 0.4084 | 0.4523 | 0.4322 |
| rDCM | 0.0000 | 0.2205 | 0.9514 | 0.7391 | 0.5860 |
| Granger Causality | 0.0000 | 0.5297 | 0.0000 | 0.3078 | 0.2648 |
| Neural Granger Causality | 0.0000 | 0.4615 | 0.0000 | 0.8235 | 0.2308 |
| CausalMamba (ours) | **0.5569** | **0.5490** | **0.8611** | **0.6941** | **0.7051** |

Table S2: Performance comparison with baseline models on 4-ROI HCP data. KPRR denotes for Known Pathway Recovery Rate.

To evaluate scalability and sensitivity to inter-subject variability, we extended our analysis to a 4-ROI visuomotor pathway, which is known to be more heterogeneous across individuals than the highly conserved visual pathway. As summarized in Table S2 and Fig. S3, **CausalMamba** recovers **56%** of the established connections (KPRR=0.56). At the link level, it achieves **F1$_{\text{pos}}$**=0.55, **F1$_{\text{neg}}$**=0.86, and **F1$_{\text{presence}}$**=0.69, yielding the best balanced score of **F1$_{\text{macro}}$**=0.71 among all methods. In contrast, rDCM exhibits a negative-edge bias (F1$_{\text{neg}}$=0.95 but F1$_{\text{pos}}$=0.22, KPRR≈ 0), Granger Causality attains moderate positives (F1$_{\text{pos}}$=0.53) but fails on negatives (F1$_{\text{neg}}$=0.00, KPRR= 0), and DCM shows lower overall performance (F1$_{\text{macro}}$=0.43, KPRR= 0). These results align with prior evidence that visuomotor networks display stable yet subject-specific connectivity patterns Mueller et al. (2013); Karahan et al. (2022), and are corroborated by our threshold-free PR analyses reported in the Supplementary. We also note that inference for our model is orders of magnitude faster; however, absolute runtimes across CPU (DCM) and GPU (DL) stacks are not directly comparable and are provided for orientation only.

## K  MODEL ARCHITECTURE ABLATION

We conducted systematic ablation studies varying both the backbone architecture and the extra neural network attached to the causality mapper to identify the optimal model configuration. Results demonstrate that Conditional Mamba, as the backbone, combined with direct utilization of its output

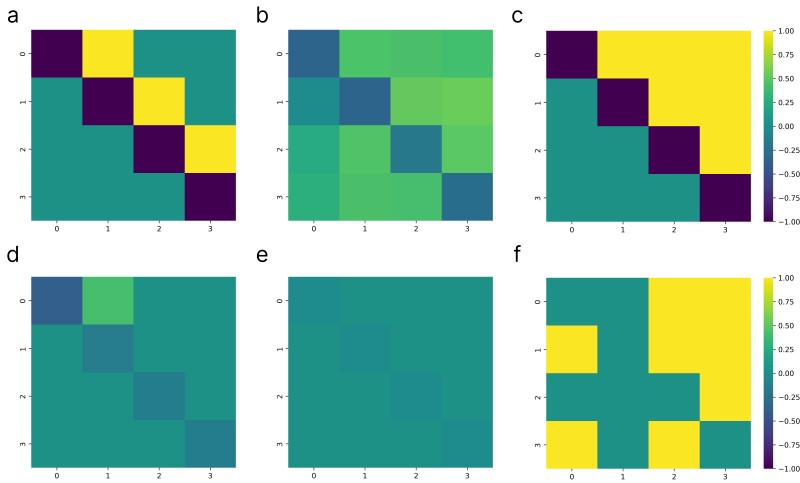

Figure S3: Coupling strengths on 4-ROI simulation data. **a** is ground truth, **b** is ours (CausalMamba), **c** is binarized matrix of ours, **d** is rDCM, **e** is DCM, **f** is Granger Causality.

features for causality prediction—without additional complex mapping modules such as GAT, GCN, and MLP—achieves superior performance. The conditional mamba backbone without extra network in the causality mapper consistently demonstrates superior performance across multiple ROI sizes (3, 4, 5, and 6).

## L USE OF LARGE LANGUAGE MODELS IN MANUSCRIPT PREPARATION

During the preparation of this manuscript, we utilized a large language model (Google's Gemini) to assist with language editing and refinement. The model's primary functions were to improve grammatical correctness, enhance clarity, and condense verbose sections for conciseness. All text generated by the model was carefully reviewed, edited, and revised by the authors to ensure that the original scientific meaning was preserved and that all claims are accurate. The conceptual framework, experimental results, and scientific conclusions presented in this paper are solely the work of the authors.

| Model | Causality Accuracy | Coupling Loss | FLOPs (M) |
|---|---|---|---|
| DCM | 0.6673 | 0.3273 | – |
| rDCM | 0.4184 | 0.3993 | – |
| Granger Causality | 0.4227 | – | – |
| Neural Granger Causality | 0.3333 | – | – |
| GRU + GAT | 0.6633±0.0078 | 0.1831±0.0042 | 69.26 |
| Conditional GRU + GAT | 0.6819±0.0055 | 0.1569±0.0023 | 96.48 |
| Mamba + GAT | 0.6445±0.0118 | 0.1831±0.0057 | 273.63 |
| Conditional Mamba + GAT | 0.6844±0.0097 | 0.1516±0.0017 | 301.28 |
| Adapter Conditional Mamba + GAT | 0.6725±0.0034 | 0.1545±0.0023 | 287.88 |
| FiLM Conditional Mamba + GAT | 0.6754±0.0075 | 0.1543±0.0024 | 274.15 |
| Hypernet Conditional Mamba + GAT | 0.6880±0.0078 | 0.1522±0.0012 | 274.28 |
| GRU + GCN | 0.6623±0.0068 | 0.1790±0.0026 | 69.18 |
| Conditional GRU + GCN | 0.6823±0.0096 | 0.1561±0.0013 | 96.40 |
| Mamba + GCN | 0.6298±0.0056 | 0.1850±0.0024 | 273.55 |
| Conditional Mamba + GCN | 0.6832±0.0056 | 0.1520±0.0019 | 301.19 |
| Adapter Conditional Mamba + GCN | 0.6867±0.0126 | 0.1514±0.0010 | 287.80 |
| FiLM Conditional Mamba + GCN | 0.6861±0.0196 | 0.1539±0.0011 | 274.07 |
| Hypernet Conditional Mamba + GCN | 0.6772±0.0173 | 0.1537±0.0023 | 274.20 |
| GRU + MLP | 0.5954±0.0352 | 0.1915±0.0052 | 78.37 |
| Conditional GRU + MLP | 0.6871±0.0085 | 0.1543±0.0012 | 105.59 |
| Mamba + MLP | 0.6538±0.0113 | 0.1873±0.0063 | 282.74 |
| Conditional Mamba + MLP | 0.6707±0.0084 | 0.1597±0.0010 | 310.38 |
| Adapter Conditional Mamba + MLP | 0.6962±0.0064 | 0.1499±0.0003 | 296.99 |
| FiLM Conditional Mamba + MLP | 0.6514±0.0511 | 0.1648±0.0123 | 283.26 |
| Hypernet Conditional Mamba + MLP | 0.6930±0.0047 | 0.1538±0.0041 | 283.39 |
| GRU | 0.6301±0.0128 | 0.1946±0.0135 | 69.16 |
| Conditional GRU | 0.6855±0.0089 | 0.1546±0.0012 | 96.37 |
| Mamba | 0.6561±0.0103 | 0.1897±0.0006 | 273.52 |
| Adapter Conditional Mamba | 0.6950±0.0051 | 0.1506±0.0004 | 287.78 |
| FiLM Conditional Mamba | 0.6885±0.0106 | 0.1536±0.0024 | 274.05 |
| Hypernet Conditional Mamba | 0.6693±0.0358 | 0.1598±0.0095 | 274.17 |
| Ours (Conditional Mamba) | **0.6904±0.0115** | 0.1517±0.0016 | 301.17 |

Table S3: Ablation study on model architecture, 3-ROIs. The performance of various combinations of backbone encoders and causality mappers are compared.

| Model | Causality Accuracy | Coupling Loss | FLOPs (M) |
|---|---|---|---|
| DCM | 0.4658 | 0.4089 | – |
| rDCM | 0.4374 | 0.4150 | – |
| Granger Causality | 0.4383 | – | – |
| Neural Granger Causality | 0.3750 | – | – |
| GRU + GAT | 0.6461±0.0046 | 0.2069±0.0016 | 92.48 |
| Conditional GRU + GAT | 0.6636±0.0035 | 0.1598±0.0036 | 128.77 |
| Mamba + GAT | 0.6379±0.0064 | 0.1910±0.0059 | 364.97 |
| Conditional Mamba + GAT | 0.6749±0.0018 | 0.1572±0.0017 | 401.83 |
| Adapter Conditional Mamba + GAT | 0.6725±0.0010 | 0.1593±0.0017 | 383.98 |
| FiLM Conditional Mamba + GAT | 0.6729±0.0036 | 0.1606±0.0048 | 365.67 |
| Hypernet Conditional Mamba + GAT | 0.6733±0.0045 | 0.1590±0.0027 | 365.84 |
| GRU + GCN | 0.6387±0.0087 | 0.2111±0.0064 | 92.37 |
| Conditional GRU + GCN | 0.6498±0.0122 | 0.1647±0.0034 | 128.66 |
| Mamba + GCN | 0.6448±0.0049 | 0.1896±0.0078 | 364.86 |
| Conditional Mamba + GCN | 0.6734±0.0015 | 0.1576±0.0020 | 401.72 |
| Adapter Conditional Mamba + GCN | 0.6700±0.0017 | 0.1603±0.0018 | 383.87 |
| FiLM Conditional Mamba + GCN | 0.6673±0.0032 | 0.1603±0.0030 | 365.56 |
| Hypernet Conditional Mamba + GCN | 0.6670±0.0020 | 0.1599±0.0020 | 365.73 |
| GRU + MLP | 0.582±0.0309 | 0.1858±0.0003 | 104.63 |
| Conditional GRU + MLP | 0.6621±0.0024 | 0.1619±0.0018 | 140.92 |
| Mamba + MLP | 0.6296±0.0033 | 0.1901±0.0035 | 377.11 |
| Conditional Mamba + MLP | 0.6645±0.0090 | 0.1623±0.0027 | 413.98 |
| Adapter Conditional Mamba + MLP | 0.6728±0.0017 | 0.1584±0.0022 | 396.12 |
| FiLM Conditional Mamba + MLP | 0.6725±0.0019 | 0.1582±0.0018 | 377.81 |
| Hypernet Conditional Mamba + MLP | 0.6734±0.0016 | 0.1580±0.0022 | 377.98 |
| GRU | 0.5377±0.0534 | 0.0951±0.0062 | 92.34 |
| Conditional GRU | 0.6588±0.0025 | 0.1620±0.0019 | 128.63 |
| Mamba | 0.6302±0.0147 | 0.1921±0.0035 | 364.82 |
| Adapter Conditional Mamba | 0.6728±0.0022 | 0.1590±0.0019 | 383.83 |
| FiLM Conditional Mamba | 0.6683±0.0022 | 0.1609±0.0032 | 365.52 |
| Hypernet Conditional Mamba | 0.6691±0.0039 | 0.1590±0.0017 | 365.70 |
| Ours (Conditional Mamba) | **0.6757±0.0042** | 0.1572±0.0024 | 401.69 |

Table S4: Ablation study on model architecture, 4-ROIs. The performance of various combinations of backbone encoders and extra neural networks attached to causality mappers are compared.

| Model | Causality Accuracy | Coupling Loss | FLOPs (M) |
|---|---|---|---|
| DCM | 0.4880 | 0.2874 | – |
| rDCM | 0.4400 | 0.2939 | – |
| Granger Causality | 0.4690 | – | – |
| Neural Granger Causality | 0.4000 | – | – |
| GRU + GAT | 0.6014±0.0042 | 0.1994±0.0045 | 115.77 |
| Conditional GRU + GAT | 0.6500±0.0042 | 0.1343±0.0003 | 161.13 |
| Mamba + GAT | 0.6076±0.0059 | 0.1690±0.0023 | 456.38 |
| Conditional Mamba + GAT | 0.6545±0.0074 | 0.1337±0.0009 | 502.46 |
| Adapter Conditional Mamba + GAT | 0.6540±0.0012 | 0.1328±0.0004 | 480.14 |
| FiLM Conditional Mamba + GAT | 0.6527±0.0073 | 0.1335±0.0007 | 457.25 |
| Hypernet Conditional Mamba + GAT | 0.6591±0.0056 | 0.1329±0.0014 | 457.47 |
| GRU + GCN | 0.5580±0.0197 | 0.1808±0.0121 | 115.63 |
| Conditional GRU + GCN | 0.6520±0.0045 | 0.1342±0.0003 | 160.99 |
| Mamba + GCN | 0.6082±0.0074 | 0.1597±0.0027 | 456.24 |
| Conditional Mamba + GCN | 0.6608±0.0079 | 0.1326±0.0017 | 502.32 |
| Adapter Conditional Mamba + GCN | 0.6561±0.0069 | 0.1341±0.0019 | 480.00 |
| FiLM Conditional Mamba + GCN | 0.6573±0.0048 | 0.1329±0.0003 | 457.11 |
| Hypernet Conditional Mamba + GCN | 0.6585±0.0062 | 0.1327±0.0008 | 457.33 |
| GRU + MLP | 0.5566±0.0013 | 0.1635±0.0009 | 130.95 |
| Conditional GRU + MLP | 0.6537±0.0044 | 0.1338±0.0002 | 176.31 |
| Mamba + MLP | 0.5603±0.0027 | 0.1600±0.0005 | 471.55 |
| Conditional Mamba + MLP | 0.6618±0.0042 | 0.1322±0.0002 | 517.63 |
| Adapter Conditional Mamba + MLP | 0.6622±0.0095 | 0.1319±0.0012 | 495.31 |
| FiLM Conditional Mamba + MLP | 0.6552±0.0018 | 0.1330±0.0002 | 472.43 |
| Hypernet Conditional Mamba + MLP | 0.6547±0.0044 | 0.1333±0.0003 | 472.64 |
| GRU | 0.5437±0.0027 | 0.1630±0.0005 | 115.59 |
| Conditional GRU | 0.6543±0.0044 | 0.1336±0.0001 | 160.94 |
| Mamba | 0.5485±0.0083 | 0.1600±0.0005 | 456.19 |
| Adapter Conditional Mamba | 0.6610±0.0016 | 0.1323±0.0002 | 457.07 |
| FiLM Conditional Mamba | 0.6586±0.0026 | 0.1326±0.0005 | 479.95 |
| Hypernet Conditional Mamba | 0.6577±0.0024 | 0.1329±0.0005 | 457.28 |
| Ours (Conditional Mamba) | **0.6704 ± 0.0041** | **0.1310±0.0006** | 502.27 |

Table S5: Ablation study on model architecture, 5-ROIs. The performance of various combinations of backbone encoders and causality mappers are compared.

| Model | Causality Accuracy | Coupling Loss | FLOPs (M) |
|---|---|---|---|
| DCM | 0.4862 | 0.2934 | – |
| rDCM | 0.4464 | 0.3002 | – |
| Granger Causality | 0.4732 | – | – |
| Neural Granger Causality | 0.4167 | – | – |
| GRU + GAT | 0.5906±0.0177 | 0.2443±0.0387 | 139.12 |
| Conditional GRU + GAT | 0.6360±0.0014 | 0.1368±0.0004 | 193.56 |
| Mamba + GAT | 0.6045±0.0016 | 0.1806±0.0057 | 547.85 |
| Conditional Mamba + GAT | 0.6416±0.0022 | 0.1361±0.0008 | 603.15 |
| Adapter Conditional Mamba + GAT | 0.6401±0.0049 | 0.1366±0.0000 | 576.36 |
| FiLM Conditional Mamba + GAT | 0.6443±0.0044 | 0.1359±0.0006 | 548.90 |
| Hypernet Conditional Mamba + GAT | 0.6407±0.0056 | 0.1361±0.0002 | 549.16 |
| GRU + GCN | 0.5756±0.0197 | 0.2013±0.0082 | 138.96 |
| Conditional GRU + GCN | 0.6354±0.0012 | 0.1368±0.0004 | 139.39 |
| Mamba + GCN | 0.6012±0.0007 | 0.1620±0.0022 | 547.68 |
| Conditional Mamba + GCN | 0.6385±0.0034 | 0.1364±0.0004 | 602.97 |
| Adapter Conditional Mamba + GCN | 0.6373±0.0002 | 0.1369±0.0003 | 576.20 |
| FiLM Conditional Mamba + GCN | 0.6393±0.0010 | 0.1364±0.0003 | 548.73 |
| Hypernet Conditional Mamba + GCN | 0.6435±0.0013 | 0.1359±0.0006 | 548.99 |
| GRU + MLP | 0.5509±0.0006 | 0.1669±0.0009 | 157.33 |
| Conditional GRU + MLP | 0.6373±0.0002 | 0.1365±0.0005 | 211.77 |
| Mamba + MLP | 0.5521±0.0020 | 0.1633±0.0010 | 566.06 |
| Conditional Mamba + MLP | 0.6439±0.0031 | 0.1357±0.0004 | 621.36 |
| Adapter Conditional Mamba + MLP | 0.6440±0.0021 | 0.1361±0.0004 | 594.57 |
| FiLM Conditional Mamba + MLP | 0.6413±0.0044 | 0.1358±0.0005 | 567.11 |
| Hypernet Conditional Mamba + MLP | 0.6423±0.0042 | 0.1358±0.0004 | 567.37 |
| GRU | 0.5431±0.0050 | 0.1663±0.0010 | 138.90 |
| Conditional GRU | 0.6406±0.0027 | 0.1363±0.0007 | 193.33 |
| Mamba | 0.5466±0.0037 | 0.1633±0.0009 | 547.63 |
| Adapter Conditional Mamba | 0.6443±0.0020 | 0.1363±0.0004 | 576.14 |
| FiLM Conditional Mamba | 0.6425±0.0026 | 0.1361±0.0007 | 548.68 |
| Hypernet Conditional Mamba | 0.6426±0.0012 | 0.1359±0.0004 | 548.94 |
| Ours (Conditional Mamba) | **0.6496±0.0040** | **0.1347±0.0010** | 602.93 |

Table S6: Ablation study on model architecture, 6-ROIs. The performance of various combinations of backbone encoders and causality mappers are compared.

