# OpenReview forum: "CAUSALMAMBA: SCALABLE CONDITIONAL STATE SPACE MODELS FOR NEURAL CAUSAL INFERENCE"
_ICLR.cc/2026/Conference — Submitted to ICLR 2026_

### Official Review · Reviewer_gyHr · 2025-10-26

**Soundness:** 2
**Presentation:** 2
**Contribution:** 3
**Rating:** 4
**Confidence:** 4

**Summary:**

This paper proposes CausalMamba, a two-stage differentiable framework for fMRI causal inference.

**Strengths:**

1. Tackles an ill-posed yet important problem, i.e., neural causality from BOLD.
2. Use of Conditional Mamba for region-specific HRF modeling.

**Weaknesses:**

1. The real-data analysis lacks a clear metric (e.g., 88% fidelity in the Abstract), and it is unclear where and how this value is calculated.
2. The “causality mapper” remains a black box, that is no interpretability or uncertainty quantification.
3. Potential data-leakage risk if simulated pretraining and HCP evaluation share HRF priors.
4. Writing overclaims general causal discovery beyond neuroimaging.

**Questions:**

1. Figure redundancy and lack of clarity
The relationship between Figure 1 (overall framework) and Figures 2-3 (module details) is poorly justified and redundant. Figure 1 already purports to illustrate the two-stage pipeline of BOLD deconvolution and causal inference, yet Figures 2 and 3 rehash the same components without adding meaningful new insights.
2. The paper fails to articulate a distinct, impactful core innovation. The two-stage approach (BOLD deconvolution + causal graph inference) is not inherently novel, as decomposing hemodynamic inverse problems has been explored in prior work (e.g., DCM variants, Granger causality adaptations, Trasfer Entropy, Attentive Transfer Entropy). The "Conditional Mamba" component is described but not positioned as a transformative advancement -- its incremental value over existing state-space models or conditional modeling techniques (e.g., FiLM, hypernetworks) is undefined. Without a clear articulation of how the framework addresses unmet needs beyond incremental engineering, the scientific contribution remains ambiguous.
3. Real-data validation lacks quantitative comparison beyond “88 % fidelity”, unclear evaluation metric.
4. Clarify whether Conditional Mamba weights are shared across ROIs or learned per-ROI.
5. The real-world validation relies on "HCP task fMRI data," but the dataset is not adequately characterized. Key details such as task design, subject demographics, ROI selection criteria, and preprocessing steps are either sparse or buried in appendices, making it impossible to assess the ecological validity of the evaluation.

---

### Official Review · Reviewer_U4qa · 2025-10-26

**Soundness:** 3
**Presentation:** 3
**Contribution:** 2
**Rating:** 4
**Confidence:** 4

**Summary:**

This paper introduces CausalMamba, a neural framework based on conditional state-space models for inferring directed causal graphs from fMRI data. By decomposing the ill-posed BOLD-to-neural inverse problem into two stages—hemodynamic deconvolution and causal graph inference—the method effectively addresses challenges related to neurovascular variability and computational complexity. The approach is validated on both simulated and real Human Connectome Project (HCP) task-fMRI data, demonstrating superior performance compared to traditional methods such as Dynamic Causal Modeling (DCM) and Granger Causality. The model shows particular promise in recovering system-level pathways and capturing task-dependent causal network reorganization. However, several issues warrant further discussion, including the reasonableness of model assumptions, comprehensiveness of evaluation, reproducibility, and handling of fundamental challenges in neural causal inference.

**Strengths:**

1. The two-stage decomposition strategy is well-motivated, breaking down the complex BOLD-to-neural inversion into learnable hemodynamic deconvolution and causal inference, thereby significantly reducing modeling difficulty.

2. The Conditional Mamba architecture effectively integrates global temporal modeling with region-specific modulation, making it suitable for capturing brain network characteristics.

3.  Experimental results show that CausalMamba outperforms all compared baselines across all metrics, and ablation studies confirm the positive contribution of each proposed module.

4. The model exhibits high computational efficiency, with complexity scaling linearly with the number of ROIs, making it suitable for whole-brain-scale analysis.

**Weaknesses:**

1. The two-stage decomposition in CausalMamba introduces strong modeling assumptions. In Stage 1, the model assumes a fixed double-gamma HRF form and a simple LFP-like neural event model. While biologically motivated, this may not capture the full diversity of real neural or hemodynamic responses. Since Stage 1 is trained solely on synthetic data, its deconvolution may not generalize well if real HRFs or neural patterns deviate from the assumed forms.

2. In Stage 2, temporal features are averaged into static node embeddings. While this simplifies the mapping, it discards precise timing information of neural events, potentially losing important cues for causal inference.

3. The "dominant direction" rule used in evaluation to resolve bidirectional predictions avoids double-counting but may discard smaller yet real reverse influences, introducing bias in the results.

4. The identifiability of HRF and neural signals is not sufficiently discussed. The model learns HRF parameters and latent event parameters by minimizing BOLD reconstruction and event losses, but different HRF-neural pairs could produce identical BOLD signals. The paper does not verify whether inferred HRFs match ground truth or known physiology.

5. The framework diagram is overly simplified and information-sparse, failing to clearly convey the model's structure and key pipeline. Its layout and aesthetics are suboptimal, undermining the clarity and readability of the core method.

6. The overall causality assumptions are problematic. The model outputs a full N×N matrix of coupling strengths and delays without imposing constraints or priors, and it lacks mechanisms to handle unobserved confounders. In fMRI, hidden brain regions and common drivers are ubiquitous, yet CausalMamba assumes the measured ROIs form a closed system. Negative couplings are interpreted as inhibition and positive as excitation, but their statistical meaning remains unclear.

7. Ablation results show only marginal differences between conditional GRU and conditional Mamba, suggesting limited incremental benefits from the proposed module. It is recommended to include a t-test and report statistical significance to determine whether the proposed method is significantly superior to competing baselines.

8. The selected baselines are largely classical and dated; the lack of comparisons with recent state-of-the-art methods diminishes empirical persuasiveness and limits the external validity of the claims.

9. The structure of the Abstract is problematic as it lacks a clear research background, making it difficult for readers to quickly grasp the paper's direction.

10. The Related Work section is relatively brief, providing neither a concise summary of existing methods nor a clear discussion of their limitations or shortcomings.

11. The experimental section uses a limited selection of baseline methods and an insufficient number of metrics to comprehensively demonstrate model effectiveness. The model was trained on a small number of datasets that are not widely adopted, undermining the credibility of the results.

12. The paper lacks descriptions of specific implementation details, hindering the reproducibility of the experimental findings.

13. Final estimates of causal connection strength lack uncertainty calibration or confidence intervals, making it difficult to distinguish reliable causal effects from spurious associations potentially driven by noise.

14. The whole-brain analysis results on 360 regions mentioned in the paper are only referenced in internal tests, without specific performance data or detailed analysis provided in the main text or appendix, weakening the persuasiveness of its scalability claims.

**Questions:**

1. How were the edge thresholds (τ+, τ−​) chosen? Were they tuned only on simulated validation data, or was any real data involved in their selection?

2. The authors report that adding graph layers degraded performance, leading to the mean-pooling design. Could you provide more insight into why temporal averaging works better? Did you try more sophisticated aggregation methods?

3. Did you evaluate how accurately Stage 1 recovers the true HRF parameters in simulation? Understanding this would clarify whether Stage 1 is fundamentally solving the deconvolution problem.

4. Do you have any preliminary results on larger synthetic networks? How do runtime and memory scale with NN and sequence length?

5. What is the purpose of the HRF Generator? Why is it necessary to obtain an HRF, and what are its specific functions?

6.  How does the ROI Index specifically select ROI-specific parameters, and how is the adapted sequence generated?

7.  Why does the causal graph obtained in Stage 2 have two parts, and what is the specific meaning of each part?

---

### Official Review · Reviewer_hfGi · 2025-10-30

**Soundness:** 2
**Presentation:** 1
**Contribution:** 2
**Rating:** 2
**Confidence:** 4

**Summary:**

The paper proposes an end-to-end deep learning approach for learning to identify causal structure in fMRI data through a process of deconvolution, feature extraction, and other networks. The model is trained on simulated data where ground truth structure is known and then applied to real data (4 regions of interest).

**Strengths:**

The problem of improving causal inference is well motivated.

**Weaknesses:**

I found the paper hard to read, imprecise, and confusing. The methodology lacks a mathematically precise description. The description and the dimensionality of variables is not sufficient to understand what is going on in Figure 1. The figure wastes a lot of white space and could have a more compact representation.  From my reading, the dimension variables are never defined in the text.
 The algorithm for conditional Mamba is very hard to parse as a commented pseudo code.

Evidence is limited because from what I can tell throughout the work there is only one model trained on simulated and tested on real data.

Compared to the main body, there is much information and detail in the appendix. From the reading the simulation was set up to follow the real data to some degree with designing the HRF and other neuroscientifically motivated choices. While I appreciate that it is still simulated data, training it to infer causal connections and then testing it on real data, seems to be a form of distant supervision. Obviously, if the simulation did not match the real as much I would expect the results to be lower. Thus, this proposed methodology does not seem to be able to be applied without strong a priori assumptions. While there has no doubt substantial effort put into the design of the simulation, the real world applications are directly tied to the simulations. Furthermore, the extensive list of alternative methods show that the proposed architecture change (which itself is hard to decipher) provides gains but not substantial. Finally, the ablation study reveals that end-to-end training already outperforms the other methods. This leads me to believe that any end-to-end training is going to overfit and the additional gains from the proposed architecture will not generalize. It is hard to tell why exactly the proposed approach works well, besides the fact that it was trained for a very similar task.
These multiple issues are intertwined and it makes it hard to see if this as a general tool for neural causal inference.

**Questions:**

Sentences are hard to parse as references are not correctly typeset. Need parenthetical references using \citep{} when reference is not part of the sentence.

Line 82 its not clear what 'modulates network behavior' means in the context. This is the biological network from which the signals are measured or the network being updated? I'm very familiar with conditional VAE and conditional GAN, but the other prior works are not given with sufficient detail to understand.  In the context of conditioning, is the ROI-specific parameters input? Or a version of the model for each ROI? This is especially important because conditional Mamba is used repeatedly Figure 1 and 2.

Line 92. It would be useful to define what 'selective mechanisms' are.

I don't know what B, N, L represent... What is the range of the ROI Index. I guess N is the ROIs? since the coupling strengths have N by N. But what is B, what is L?

In Figure 1a is the HRF same size as the deconvolved BOLD?

In Figure 2 are the conditional Mamba the same in each parallel stream?

In Figure 3 what does 'expand & concat' do?


Why is training on simulated data expected to generalize to real data? There seems to be potential for large gap.

Looking at Table 2 it appears the first strategy that maps "BOLD to Causality" outperforms all baselines. Does this mean that end-to-end training without CausalMamba is sufficient to outperform the baselines?

In Table S2, the proposed method is outperformed in terms of F1 presence by both rDCM and Neural Granger Causality. This is not discussed. Accurately detecting presence of a connection would seem to be the primary task, and should temper the claims.


Line 224 "wo orders"

---

### Official Review · Reviewer_g1K8 · 2025-11-01

**Soundness:** 1
**Presentation:** 2
**Contribution:** 1
**Rating:** 2
**Confidence:** 2

**Summary:**

The paper introduces a scalable algorithm to recover effective connectivity of fMRI BOLD data based on the MAMBA structure.

**Strengths:**

a nice use of MAMBA

**Weaknesses:**

The authors would like to report that we could use MAMBA to do causal inference on fMRI BOLD data. Although this would have been clear, it is nice to see that it works. However, the authors claim that this is a massive progress in the field because it is scalable and works better than the DCM family.  This would have been a great method in 2020 (though MAMBA was not there yet), but now we have much better methodologies that do the same thing, and I suspect they will outperform the technique proposal here by far. Basically, what I am saying here, you need to compare your technique to the state of the art to say that it is better than the others or altentaively show theoretical or empirical evidence to show that it succeeds where the others fail, even if it is not always the most accurate.

Concrete, I expect the authors to compare their technique at least with the following two methods and show they are superior, otherwise I don't think there :
1) Luo, Z., Peng, K., Liang, Z., Cai, S., Xu, C., Li, D., ... & Liu, Q. (2025). Mapping effective connectivity by virtually perturbing a surrogate brain. Nature Methods, 1-10.
2) Arab, F., Ghassami, A., Jamalabadi, H., Peters, M. A., & Nozari, E. (2025). Whole-brain causal discovery using fMRI. Network Neuroscience, 9(1), 392-420.

**Questions:**

see above

---

### Meta-Review · Area_Chair_kNsz · 2026-01-03

**Summary:**

This paper proposes CausalMamba, a two-stage, differentiable framework for causal inference from fMRI data. The method decomposes the ill-posed problem of inferring neural causality from BOLD signals into (1) hemodynamic deconvolution to recover latent neural activity and (2) causal graph inference using a Conditional Mamba state-space architecture. The authors report improved scalability and performance relative to classical baselines such as Dynamic Causal Modeling (DCM), and present results on both simulated data and real task fMRI datasets, including claims of recovering canonical neural pathways and task-dependent network reconfiguration.

Reviewers broadly agreed that the problem is important and that the proposed approach is technically interesting, particularly in its scalability and use of modern sequence models. However, there was substantial disagreement regarding the novelty, clarity, empirical rigor, and strength of causal claims, leading to mixed but overall negative evaluations. The primary concerns centered on incomplete comparisons with recent state-of-the-art methods, strong and insufficiently justified modeling assumptions, limited and potentially confounded real-data validation, and unclear articulation of the core scientific contribution beyond incremental architectural engineering.

**Reviewer Concerns:**

There is no rebuttal.

**Reviewer Scores:**

There is no rebuttal nor discussion.

---

### Decision · Program_Chairs · 2026-01-26

Reject